# Eyes Tell the Truth: GazeVal Highlights Shortcomings of Generative AI in Medical Imaging

David Wong[1]*, Bin Wang[1]*, Gorkem Durak[1], Marouane Tliba[2], Akshay Chaudhari[3],
Aladine Chetouani[4], Ahmet Enis Cetin[2], Cagdas Topel[1], Nicolo Gennaro[1], Camila Lopes Vendrami[1],
Tugce Agirlar Trabzonlu[1], Amir Ali Rahsepar[1], Laetitia Perronne[1], Matthew Antalek[1], Onural Ozturk[1],
Gokcan Okur[5], Andrew C. Gordon[1], Ayis Pyrros[6], Frank H. Miller[1], Amir Borhani[1], Hatice Savas[1],
Eric Hart[1], Drew Torigian[7], Jayaram K. Udupa[7], Elizabeth Krupinski[8], Ulas Bagci[1]
Northwestern University[1], University of Illinois at Chicago[2], Stanford University[3],
Université Sorbonne Paris Nord[4], Loyola University Chicago[5],
DuPage Medical Group[6], University of Pennsylvania[7], Emory University[8]
ulas.bagci@northwestern.edu

## Abstract

*The demand for high-quality synthetic data for model training and augmentation has never been greater in medical imaging. However, current evaluations predominantly rely on computational metrics that fail to align with human expert recognition. This leads to synthetic images that may appear realistic numerically but lack clinical authenticity, posing significant challenges in ensuring the reliability and effectiveness of AI-driven medical tools. To address this gap, we introduce **GazeVal**, a practical framework that synergizes expert eye-tracking data with direct radiological evaluations to assess the quality of synthetic medical images. **GazeVal** leverages gaze patterns of radiologists as they provide a deeper understanding of how experts perceive and interact with synthetic data in different tasks (i.e., diagnostic or Turing tests). Experiments with sixteen radiologists revealed that 96.6% of the generated images (by the most recent state-of-the-art AI algorithm) were identified as fake, demonstrating the limitations of generative AI in producing clinically accurate images.*

## 1. Introduction

Advances in machine learning and artificial intelligence (AI) have revolutionized many fields, including medical imaging, offering new horizons in diagnostics, prognostics, and personalized medicine. Central to these advancements is the availability of large-scale, high-quality datasets [4]. The significance of expansive data repositories in medical imaging cannot be overstated. Despite its importance, as-sembling large-scale medical imaging datasets faces several obstacles, including privacy concerns, data annotation costs and resources, and data imbalance and bias issues. Synthetic data generation has emerged as a vital tool to mitigate the challenges associated with acquiring large-scale medical imaging data. By using techniques such as generative adversarial networks (GANs) [5], variational autoencoders (VAEs) [18] [13], and most recently diffusion probabilistic generative algorithms [7], it is possible to create realistic medical images that can augment existing datasets.

The integration of synthetic data into medical imaging is becoming an essential method for training robust machine learning models, enhancing data augmentation, balancing classes, preserving patient privacy, simulating rare conditions, and so on. However, current evaluation methodologies for assessing the quality of synthetic images predominantly rely on computational metrics that often fail to align with human expert recognition, resulting in synthetic images that may appear numerically realistic yet lack clinical authenticity. This discrepancy undermines the reliability, generalizability, and clinical utility of AI-driven medical tools.

As deep generative models increasingly rely on synthetic data, it is essential to investigate the quality of the generated images. While traditional benchmarks often focus on evaluating a model's ability to generate realistic images that accurately represent the given condition, it is equally critical to assess whether generated images contain unrealistic regions or discriminative features that diverge from real-world images. We hypothesize that discriminative features within generated images are including (but not limited to):

- **Artifacts:** Unnatural patterns or anomalies that arise from the synthesis process.

---

*Equal contribution.

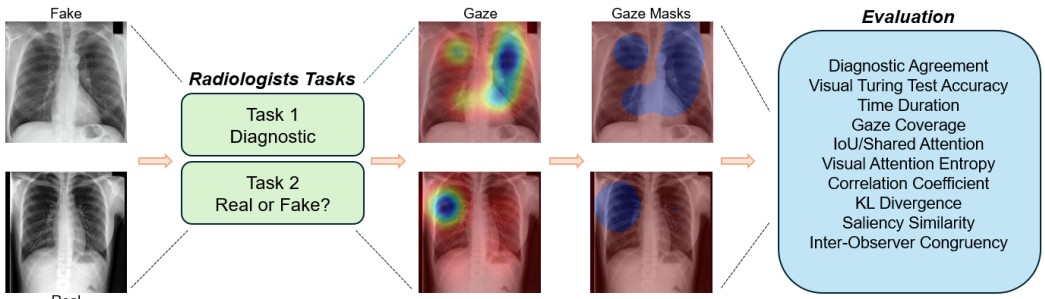

Figure 1. Overview of the proposed GazeVal framework, which introduces two tasks with corresponding evaluation metrics to quantitatively assess the quality of synthetic Chest X-ray images with expert knowledge.

- **Over- or under-exposure:** Regions with excessive brightness or darkness that depart from the expected representativeness of the image.
- **Texture and pattern inconsistencies:** Discrepancies in texture or patterns that deviate from the expected visual semantics.

These features can be overlooked when evaluating a model's ability to generate realistic images. However, they represent critical errors that can undermine the diagnostic accuracy and efficacy of models trained on synthetic data. The presence of unrealistic images in deep generative models can have severe consequences, including (1) reduced diagnostic accuracy: unrealistic medical images can lead to incorrect diagnoses or classifications, ultimately compromising the reliability of medical imaging analysis, and (2) decreased model performance: the presence of unrealistic images can also degrade the overall performance of deep generative models, as they struggle to generalize effectively to real-world data [11].

Towards this, we propose to quantify realism and evaluate discriminative imaging features together. Generative algorithms use computational metrics to assess the level of realism in generated images, but they do not often include Visual Turing Test (VTT) to explore high-level human perception. Assessing the presence and persistence of discriminative features in generated images is also important from the human recognition point of view. Specifically in this study, we are interested in exploring when and why radiologists are concerned about the realism of generated images. To accomplish this, we introduce **GazeVal**, a practical framework that synergizes expert eye-tracking data with direct radiological evaluations of synthetic medical images generated by the recent state-of-the-art diffusion generative algorithms (Figure 1). We aimed to understand how experts perceive and interact with synthetic data in two different tasks, diagnostic and Visual Turing tasks, with sixteen radiologists as participants.

## 2. Related Works

### 2.1. Generative Models

Among current generative models, Latent Diffusion Models (LDMs), including DALL-E 2 [17], Imagen [20], Stable Diffusion [19], and RoentGen [1], are notable for their ability to produce high-quality images. These models leverage conditional inputs, such as text prompts or image embeddings, to guide the denoising process from a noise-initialized image [19]. A key advantage of LDMs lies in their ability to generate high-resolution images quickly and stably, addressing issues faced by previous models. Variational Autoencoders (VAEs) [12], for instance, struggled with the quality of high-resolution outputs, while Generative Adversarial Networks (GANs) [6] often encountered training instabilities. In contrast, LDMs provide consistent, high-quality image generation with improved training stability, making them highly effective for high-resolution applications. Despite these advancements, current evaluation methods for LDMs still rely heavily on computational metrics, which do not always align with human expert assessment. Consequently, some images that score well on quantitative metrics may lack clinical realism, posing challenges for AI-based medical applications. Our innovative **GazeVal** framework combines expert eye-tracking data with radiologist evaluations to more accurately assess the clinical authenticity of synthetic medical images, ensuring their reliability and effectiveness in medical applications.

### 2.2. Visual Turing Test

The Visual Turing Test (VTT) is a method used to evaluate the realism and quality of synthetic or AI-processed images by determining whether they are indistinguishable from real images to human observers, particularly experts like radiologists viewing medical images [2]. Inspired by the original Turing Test [22], which assesses a machine's ability to exhibit human-like intelligence, the VTT in this context gauges whether AI can produce medical images that are perceptually and diagnostically equivalent to those ac-

quired from actual patients. This method has become increasingly popular for assessing GANs' capability to produce high-quality images that closely resemble real ones. For instance, Chuquicusma et al. used VTT to examine the realism of GAN-generated CT scans of lung nodules [2], while Hong et al. applied it to synthetic lumbar MR images generated from CT axial inputs [9]. Myong et al. leveraged VTT to confirm the authenticity of GAN-generated chest radiographs [15], and Park et al. [16] validated the quality of GAN-produced body CT scans through VTT. While VTT has been extensively used in evaluating GANs, its application to LDMs remains unexplored. This study extends the utility of VTTs to the evaluation of generative models in medical imaging, specifically focusing on LDMs, which represent the current state-of-the-art generative models.

## 3. Methods

The framework of our proposed **GazeVal** is illustrated in Figure 1. Given a real chest X-ray image and its associated report, a synthetic chest X-ray image is generated based on the report by an LDM-based generative model, RoentGen (Sec. 3.1). The synthetic images are then placed in a dataset with real images and reviewed by radiologists under two task settings (Sec. 3.2.1). First, radiologists are asked to provide a diagnosis without knowing that synthetic images are included. Second, they are asked to determine whether each image is real or generated. At the same time, we use eye tracking to record their eye gaze during the tasks (Sec. 3.2.2). Using the gaze data and their task answers, we can compare synthetic and real X-rays across various metrics to evaluate the quality of the synthetic images.

### 3.1. Synthetic Chest X-ray Generation

We first generate synthetic chest X-rays using RoentGen [1], an LDM-based generative model. Its model architecture is demonstrated in Figure 2. A noisy image vector is conditionally denoised by a U-Net and a text embedding generated by the encoder. The variational autoencoder then decodes the latent vector and maps it to a pixel space, which results in a high-resolution generated image. To create synthetic images conditionally, we utilized patient reports from the MIMIC-CXR dataset [10]. Each report corresponds to a real chest X-ray image. To avoid unintentional loss of information, each selected report is required to meet RoentGen's token length limit. These reports were then processed through RoentGen, producing synthetic chest X-ray images at a resolution of 512x512 with 75 inference steps. This method of generation ensures each synthetic chest X-ray has a corresponding real X-ray with the same report content. Then, an independent radiologist reviewed the generated images, filtering out those with significant unrealistic features. Thirty images were randomly selected from the filtered set for use in the experiment. Some examples of the

synthetic chest X-ray images are displayed in Figure 3.

### 3.2. GazeVal

#### 3.2.1. Diagnostic and Visual Turing Test Task

The synthetic chest X-ray images are randomly mixed with real images and reviewed by radiologists under two distinct tasks: a diagnostic assessment and a VTT. In the first task, radiologists are asked to diagnose each chest X-ray image by verbally describing any pathologies or findings they observe. Importantly, they are not informed that some of the images are synthetic. If a synthetic image appears too unrealistic or contains incorrect pathologies, the radiologist might detect these quality issues naturally through their assessment. In the second task, radiologists are explicitly informed that the dataset includes both real and synthetic chest X-ray images. Their task is then to identify each image as either real or synthetic. This differs from the first task, where radiologists were not informed about the presence of synthetic images. By directly guiding radiologists to consider the possibility of synthetic images, this approach enables a more direct assessment of image quality. Afterward, a voting system will be used to allow the radiologists to collaborate without interaction, where each radiologist gets one vote regarding the type of image being observed. Notice that an average interval of ten days is set between the two tasks to reduce potential bias from prior exposure to the images.

#### 3.2.2. Eye Tracking Setup

We used eye tracking device to capture radiologists' search patterns so that we can know whether their visual attention is different between real and synthetic chest X-ray images. In this study, we used the EyeLink 1000 Plus device which uses a single camera at 500 Hz to track eye movements for both pupils. The gaze point is determined as the average of the two pupils. The eye gazes of the radiologists were collected with a fixation being defined as an area where saccades have a velocity of less than 30°/s and acceleration less than $8000/s^2$ [3]. This tracking is enabled by a 13-point calibration process before each experiment, which was validated within the EyeLink calibration system. During the experiment, this system is also used to correct for blinking and other artifacts, as well as account for head movements.

#### 3.2.3. Gaze Masking

The raw gaze data are normally transformed into visual attention maps to qualitatively represent the eye movements of the users [23]. However, for this study, we require a quantitative comparison of eye gaze patterns, which visual attention maps cannot provide. Therefore, we introduce a gaze masking technique to facilitate quantitative gaze analysis. This involves applying a threshold to the attention map, as shown in Figure 4. Although the mask may encompass areas the radiologist did not actively examine, it effec-

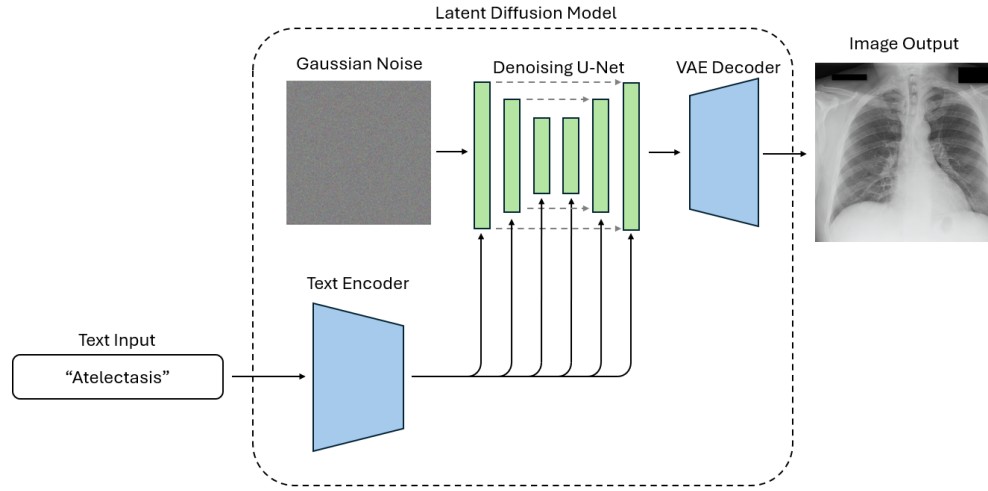

Figure 2. The pipeline of generating synthetic chest X-ray images.

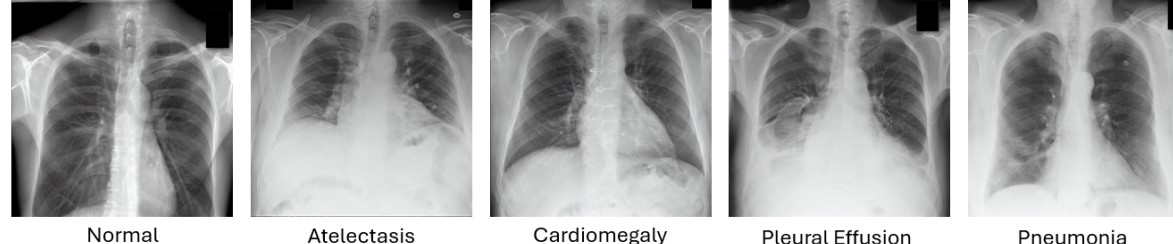

Figure 3. Synthetic X-rays generated by RoentGen with reports from MIMIC-CXR. Each X-ray is labeled with important features mentioned in the reports.

tively represents the overall regions observed. This masking approach enables us to conduct quantitative measurements, such as gaze coverage and intersection over union metric.

### 3.3. Experimental Setting

In our experiment, 30 *generated* chest X-ray images and 30 *real* chest X-ray images are randomly mixed to create a dataset of 60 images. The image sequence remains constant across experiments with **16 different radiologists**. As illustrated in Figure 5, the images are displayed at the center of the monitor. The eye tracker is placed 30 cm in front of the monitor, and the distance between the radiologist and the monitor is set as 80cm during calibration.

### 3.4. Radiologist Expertise and Experience

The participating radiologists is comprised of 5 women and 11 men. The distribution of years of experience is as follows: 2 individuals with 0-4 years of experience, 6 have 5-9 years, 5 have 10-19 years, and 3 have over 20 years of experience. The group also includes various specialties including: body imaging, cardiothoracic radiology, interventional radiology, neuroradiology, and musculoskeletal radiology. This broken down in Table 1.

Table 1. Radiologists listed by specialty and years of experience. Radiologists with multiple subspecialities are listed multiple times.

| Specialty | Years of Experience | | | |
|---|---|---|---|---|
| | 0-9 | 10-19 | 20+ | Total |
| Body Imaging | 3 | 3 | 2 | 8 |
| Cardiothoracic Imaging | 4 | 2 | 1 | 7 |
| Musculoskeletal Imaging | 1 | 0 | 1 | 2 |
| Neuroradiology | 0 | 0 | 1 | 1 |
| Interventional Radiology | 1 | 1 | 0 | 2 |

## 4. Results

To better evaluate the quality of synthetic images and their difference with real images, we propose the following evaluation metrics:

- **Diagnostic Agreement**: the percentage of cases in which the radiologists' verbal reports align with the original image reports (gold standard). An aligned verbal report is a report that contains any pathologies found in the original image reports. This reflects the model's ability to reproduce pathologies.

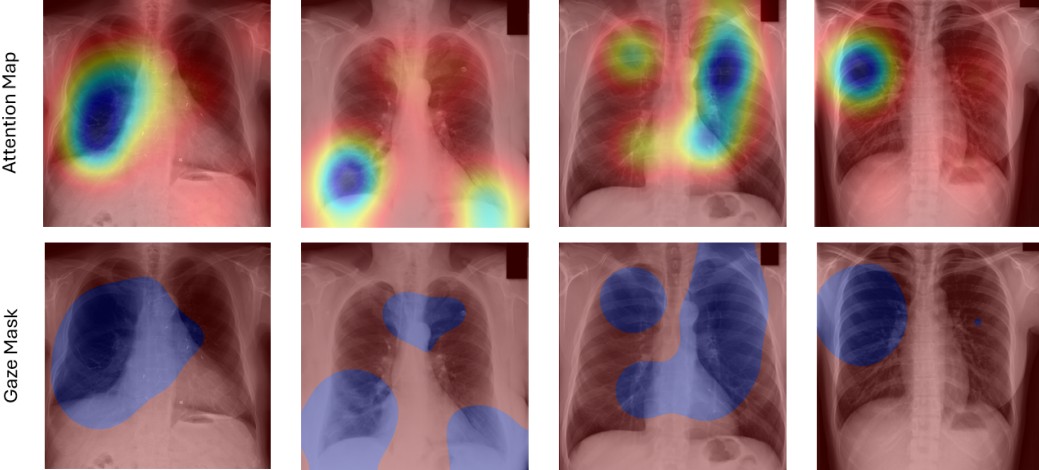

Figure 4. The top row contains samples of the attention maps produced from the eye gazes of radiologists. The bottom row contains their related gaze masks.

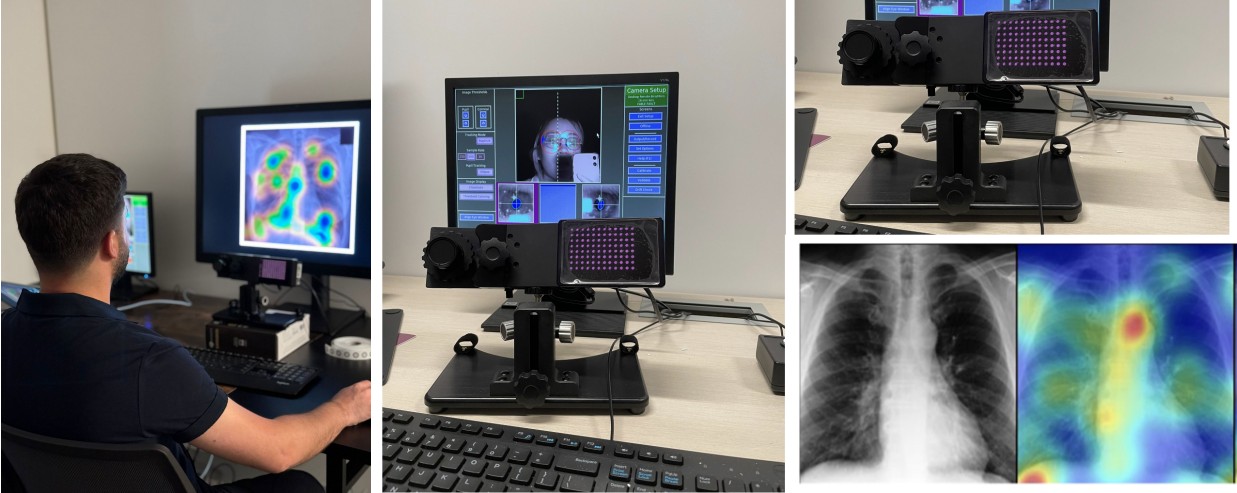

Figure 5. Left: Eye tracking setup. The radiologist is viewing the image on the monitor with the eye tracker in between them. Middle: EyeLink 1000 Plus eye-tracker view with calibration software. Right: Eye-tracker and example attention map.

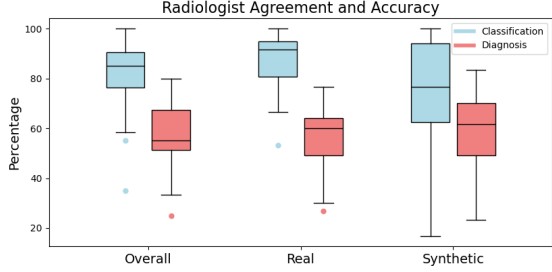

Figure 6. The plot shows the real-fake classification accuracy in blue and the diagnosis agreement in red for real X-rays, synthetic X-rays, and both combined.

- **VTT Accuracy**: the percentage of cases in which the radiologist can correctly identify the category of the image (real or synthetic). This metric reveals how good the image quality is to fool the radiologist.
- **Time Duration**: the average time that the radiologist spends on reading an X-ray.
- **Gaze Coverage**: the percentage of the image that the radiologist's eye gaze covers.
- **Intersection over Union (IoU)**: the percentage of overlap between two gaze masks, calculated by the area of their intersection divided by the area of their union.
- **Visual Attention Entropy**: the average entropy of each radiologist's visual attention for real and synthetic X-rays was compared with a paired t-test.

- **Correlation Coefficient (CC)**: measures the strength and direction of the linear relationship between viewing patterns of real and synthetic X-rays. Higher values indicate stronger alignment, particularly for the first and last fixations, while lower values for the longest and shortest fixations suggest weaker alignment.
- **KL Divergence (KLD)**: quantifies the difference between saliency distributions. Lower values indicate higher similarity, especially in the first and last fixations, while larger differences appear in the shortest and longest fixations.
- **Saliency Similarity (SS)**: evaluates the overall similarity between saliency maps. Higher values suggest greater alignment, particularly in the first and last fixations, while the longest and shortest fixations show weaker similarity.
- **Inter-Observer Congruency (IOC)**: Quantifies the consistency of visual attention across different observers. *Fixation-Based IOC* evaluates spatial congruency by comparing each observer's fixation points to a binary saliency mask generated from the fixations of all other observers. *Scanpath-Based IOC* assesses both spatial and temporal agreement in gaze sequences; mainly, it provides insight into the consistency of gaze transitions across observers.

### 4.1. Synthetic X-rays Closely Mimic Real X-rays in Representing Diseases

From Table 2, there is an average 55% and 59% diagnostic agreement between the radiologists and the reports for real and synthetic X-rays, respectively. This level of agreement is expected with regards to previous studies investigating diagnostic accuracy when studying chest X-rays [8, 21]. This means the diagnostic agreement for real and synthetic images are similar. As illustrated in Figure 6, there is no significant difference in diagnostic agreement when interpreting real and synthetic X-rays based upon a paired t-test with $p = 0.17$. A similar conclusion is drawn when comparing the average visual attention entropy between real and synthetic X-rays during the diagnosis task, with a paired t-test yielding a p-value of 0.91. These results suggest that synthetic X-rays represent diseases similarly to real X-rays.

### 4.2. Viewing Patterns Differ on Synthetic X-rays Compared to Real X-rays

**Temporal Fixation Analysis:** Table 3 compares viewing patterns on real and synthetic X-rays using saliency-based metrics (CC, KLD, and SIM). The results indicate stronger alignment in first and last fixations, suggesting that initial and final attention patterns remain consistent across real and synthetic images. However, the longest and shortest fixations show greater divergence, implying that radiologists exhibit more variability in extended viewing behavior when assessing synthetic images. This suggests that while early and concluding fixations remain stable, differences emerge during prolonged visual processing.

**IOC Analysis:** Table 4 presents the IOC results for real and AI-generated X-ray images across two tasks, using Fixation-Based and Scanpath-Based Congruency (DTW and Levenshtein) [14]. Fixation-Based IOC measures spatial alignment in fixation density maps, while Scanpath-Based IOC evaluates gaze trajectory similarity using DTW and Levenshtein distance.

Fixation-Based IOC results show that real X-rays exhibit higher congruency than synthetic ones in both tasks. In Task 1, mean IOC scores are 0.6920 for real images and 0.6695 for synthetic ones, while Task 2 shows lower values (0.5272 and 0.5130, respectively), indicating increased variability in gaze distribution, potentially due to task complexity. Scanpath-Based IOC follows a distinct trend, with Task 1 showing lower congruency than Task 2 across both DTW (0.3351 vs. 0.4409 for real; 0.3198 vs. 0.4362 for synthetic) and Levenshtein measures (0.3117 vs. 0.4166 for real; 0.3059 vs. 0.3889 for synthetic), suggesting that gaze sequence consistency improves in Task 2.

Standard deviation values further highlight variability in observer agreement. Fixation-Based IOC shows greater variation in Task 2 (Std Dev: 0.1530 real, 0.1581 synthetic) than Task 1 (0.1171, 0.1245), indicating more divergence in viewing patterns. Similarly, Scanpath-Based IOC reveals higher variability in Task 2, with Levenshtein scores (0.2366 real, 0.2423 synthetic) showing less stability across observers. These findings suggest that radiologists adapt distinct visual search strategies when interpreting synthetic images, influencing diagnostic consistency.

### 4.3. Radiologists Can Still Easily Identify Real or Synthetic

As shown in Table 2 and Figure 6, radiologists correctly identified 86% of real images and 74% of synthetic images. Across all X-rays, the radiologists achieved an average specificity of 79.2% and an average sensitivity of 81.1% with a standard deviation of 18.25% and 20.0%, respectively, where a positive case is defined as a case that is classified as a synthetic X-ray. When collaborating through a voting process, their accuracy in determining whether an X-ray was real or synthetic increased to 96.7%. This demonstrates that, in almost all cases, radiologists can accurately distinguish between real and synthetic images, which means the current state-of-the-art generative model are unable to generate highly realistic radiological images yet. From the analysis based on their gaze mask, we find that the most difficult image components to model are bone structure, medical devices, and small details, such as lung parenchyma and vascular anatomy, as shown in Figure 7.

Table 2. Consolidated view of time duration, VTT accuracy, and gaze coverage for each task and X-ray type.

| Task + X-ray Type | Time Duration (s) | VTT Accuracy | Diagnostic Agreement | Gaze Coverage |
|---|---|---|---|---|
| Diagnosis + Real | $28.21 \pm 19.84$ | - | $0.55 \pm 0.13$ | $0.86 \pm 0.12$ |
| Diagnosis + Synthetic | $30.28 \pm 21.56$ | - | $0.59 \pm 0.17$ | $0.87 \pm 0.13$ |
| VTT + Real | $8.38 \pm 7.71$ | $0.86 \pm 0.13$ | - | $0.76 \pm 0.20$ |
| VTT + Synthetic | $8.74 \pm 8.83$ | $0.74 \pm 0.24$ | - | $0.76 \pm 0.19$ |

Table 3. Saliency Metrics Comparing Real and Fake Bias Maps for Task 1 and Task 2.

| Condition | Task 1 | | | Task 2 | | |
|---|---|---|---|---|---|---|
| | CC | KLD | SIM | CC | KLD | SIM |
| First | 0.4706 | 6.0931 | 0.4327 | 0.5158 | 7.0446 | 0.4286 |
| Last | 0.4125 | 6.8897 | 0.3995 | 0.4777 | 6.8679 | 0.4104 |
| Longest | 0.1719 | 8.1881 | 0.3130 | 0.1938 | 9.7120 | 0.2839 |
| Shortest | 0.1877 | 8.6826 | 0.3127 | 0.2044 | 10.2255 | 0.2860 |

## 4.4. Generative Model Struggles to Generate Realistic X-rays with Pathologic Findings

As illustrated in Table 5, the VTT accuracy for synthetic normal chest X-rays (without pathologies) is notably lower than that for synthetic chest X-rays containing pathologies. This suggests that radiologists find it easier to identify synthetic images when they contain specific pathologies, such as cardiomegaly, pleural effusions, atelectasis, and pneumonia. Consequently, generating realistic synthetic X-rays with pathologies presents a greater challenge than producing synthetic X-rays without any pathologies.

## 4.5. Human Visual Attention is Task-Guided

Another key finding is that human visual attention can be affected when assigned different tasks. We define shared attention, which represents the overlap between gaze masks from the diagnostic task and the VTT, as illustrated in Figure 8. To better quantify this, the intersection over the union (IoU) of the two masks is calculated as the shared attention. In our experiment, we observed that the shared attention between the two tasks was 73%, which is relatively low, as a much higher IoU value is typically necessary to ensure good alignment. This suggests that radiologists' attention shifts between tasks, even when reading the same image. This gives us the insight that human visual attention can be substantially altered when given varying tasks.

This finding can also be supported by measuring gaze time duration and gaze coverage. As shown in Table 2, the average duration of the diagnostic task is around 30 seconds while the VTT task only takes about 8 seconds. For the gaze coverage, around 86% of the image are covered by the radiologists' gaze attention during the diagnostic task but for the VTT, it is 76%. These variations in time duration and gaze coverage further indicate that human visual attention can shift significantly across different tasks.

## 5. Discussion

In this study, we evaluate the clinical authenticity of synthetic X-rays generated by LDMs. Our findings show that radiologists can distinguish between real and generated images with high accuracy, especially when collaborating. While the diagnostic agreement between real and synthetic images suggests that LDMs can generate plausible pathologies, challenges remain in producing realistic representations of complex conditions. Results from the VTT further indicate that LDMs require further refinement to enhance image fidelity.

Beyond diagnostic accuracy, we also examine how radiologists' gaze behavior varies across tasks. Comparing gaze masks between the diagnostic and VTT tasks using the IoU metric highlights the task-driven nature of human visual attention. While a substantial portion of attention overlaps between tasks, each task also prompts the exploration of new areas. This pattern is consistent across both real and synthetic images, with no significant differences observed in other metrics, such as gaze coverage or time spent per image. These findings suggest that shifts in visual attention are primarily guided by the task rather than the image itself. This has important implications for future studies comparing human and machine attention, emphasizing the need to account for both the task assigned to humans and the prompt given to AI models. Moreover, it raises the question of whether changing prompts for language-vision models similarly influences spatial attention, an area ripe for further investigation. Understanding these task-driven variations could also help AI systems better inform or guide human attention during clinical assessments.

While our study focuses on synthetic X-rays, the evaluation of generative models for CT and MRI images remains unexplored. Extending GazeVal to synthetic 3D medical images could provide valuable insights into model performance across different imaging modalities. Additionally, the study design, which includes 60 X-rays per session, may

Table 4. Comparison of Inter-Observer Congruency (IOC) in Task 1 and Task 2 using Fixation-Based Congruency and Scanpath-Based Congruency (DTW and Levenshtein).

| Method | Type | Task 1 | | | | | | | Task 2 | | | | | | |
|---|---|---|---|---|---|---|---|---|---|---|---|---|---|---|---|
| | | Mean | Min | Max | Median | Std Dev | 25% | 75% | Mean | Min | Max | Median | Std Dev | 25% | 75% |
| Fixation | Real | 0.6920 | 0.4366 | 0.9068 | 0.6985 | 0.1171 | 0.6295 | 0.7723 | 0.5272 | 0.2513 | 0.8305 | 0.5101 | 0.1530 | 0.4322 | 0.6156 |
| | Fake | 0.6695 | 0.3813 | 0.8903 | 0.6722 | 0.1245 | 0.5936 | 0.7560 | 0.5130 | 0.2359 | 0.8625 | 0.4980 | 0.1581 | 0.4133 | 0.5991 |
| DTW | Real | 0.3351 | 0.0000 | 1.0000 | 0.3107 | 0.1720 | 0.2191 | 0.4170 | 0.4409 | 0.0000 | 1.0000 | 0.4242 | 0.1972 | 0.3201 | 0.5317 |
| | Fake | 0.3198 | 0.0000 | 1.0000 | 0.2869 | 0.1745 | 0.2029 | 0.3963 | 0.4362 | 0.0000 | 1.0000 | 0.4232 | 0.1906 | 0.3089 | 0.5469 |
| Levenshtein | Real | 0.3117 | 0.0000 | 1.0000 | 0.2390 | 0.2294 | 0.1346 | 0.4361 | 0.4166 | 0.0000 | 1.0000 | 0.3636 | 0.2366 | 0.2258 | 0.5786 |
| | Fake | 0.3059 | 0.0000 | 1.0000 | 0.2258 | 0.2328 | 0.1256 | 0.4400 | 0.3889 | 0.0000 | 1.0000 | 0.3292 | 0.2423 | 0.1948 | 0.5521 |

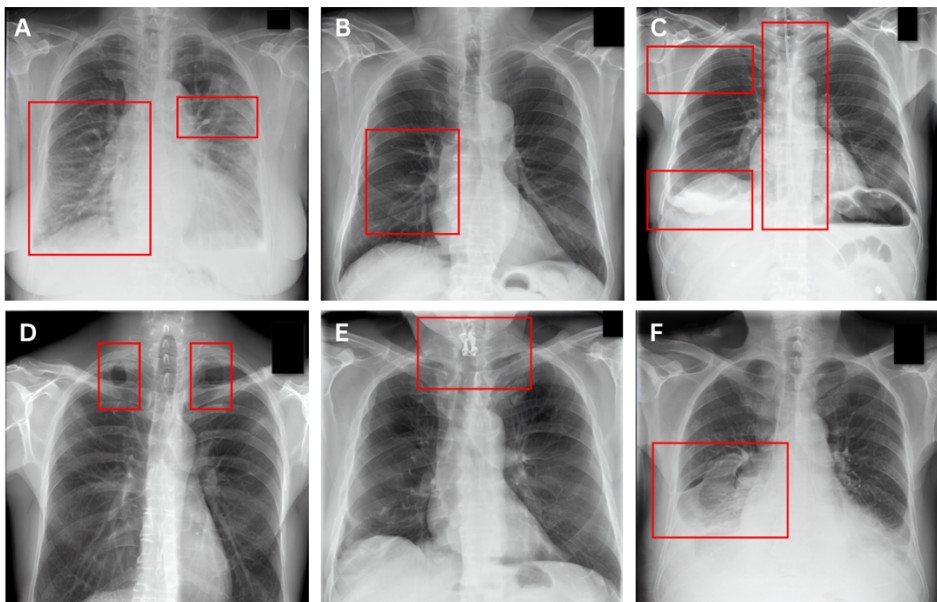

Figure 7. These X-rays have red bounding boxes indicating the unrealistic parts. (a) unrealistic bone anatomy. (b) inconsistent vascular anatomy and interrupted pulmonary artery branches. (c) unrealistic medical devices and excessive plasticity. (d) areas of inconsistent air density (e, f) unrealistic anatomical and pathological structures.

Table 5. Comparison of VTT accuracy of synthetic chest X-ray images without pathology (normal) and with different pathologies.

| Pathology | VTT Accuracy (Synthetic) |
|---|---|
| Normal | **0.7539** |
| Cardiomegaly | 0.8125 |
| Pleural Effusion | 0.8750 |
| Pneumonia | 0.7500 |

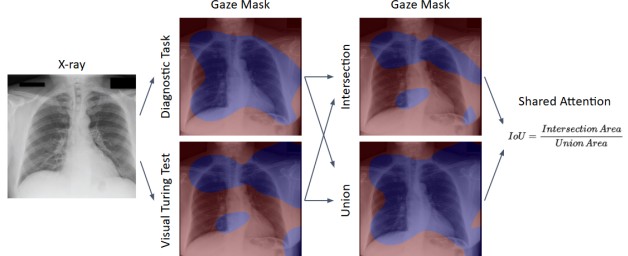

Figure 8. The illustration of shared attention calculation.

introduce fatigue and situational biases, as this workload exceeds typical radiologist reading sessions. Prolonged exposure could affect visual attention and decision-making, suggesting that future studies should consider optimizing session length to better reflect real-world clinical conditions.

## 6. Conclusion

**GazeVal** demonstrates a significant advancement over conventional computational metrics by incorporating human expertise into the evaluation process. This approach not only highlights the shortcomings of current generative models but also guides the development of more clinically relevant synthetic data. By ensuring that synthetic datasets meet the stringent standards required for real-world medical applications, **GazeVal** paves the way for more trustworthy and effective AI solutions in healthcare, ultimately enhancing diagnostic accuracy and patient outcomes.

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
