# OpenReview forum: "Eye Tell the Truth: GazeVal Highlights Shortcomings of Generative AI in Medical Imaging"
_thecvf.com/CVPR/2025/Workshop/SyntaGen — SyntaGen 2025 Poster_

### Official Review · Reviewer_W9yB · 2025-03-24

**Rating:** 6
**Confidence:** 4

**Review:**

While existing computational metrics fall short in generating X-ray images with clinical authenticity, this paper proposes leveraging expert eye-tracking data combined with direct radiological evaluations. The quality of this submission is **above average** for a workshop paper, and the investigated topic is relevant with synthetic data.

The overall presentation is clear. In terms of significance, I appreciate the authors’ effort in incorporating a diverse set of evaluation metrics grounded in expert knowledge. The paper also offers several valuable insights into the limitations of synthetic X-ray images generated by state-of-the-art latent diffusion models (LDMs). There are, however, a few minor suggestions to improve clarity:
(1) Figure 2 occupies a substantial amount of space but primarily illustrates that a text-to-image diffusion model is used—consider simplifying into plain text or resizing it;
(2) The subsections in the experimental analysis section can follow the table numbering to provide better readability.

One of the most significant contributions of AI-driven medical tools is their ability to automate clinical workflows while safeguarding patient privacy. In this regard, I encourage the authors to explore approaches such as human preference-guided optimization [1] of LDMs to generate more realistic and high-fidelity medical images. This direction holds great promise and could make meaningful contributions to both the academic and medical communities.

*References*
[1] Rich human feedback for text-to-image generation. Liang et al. CVPR 2024.

---

### Official Review · Reviewer_odL9 · 2025-03-25
**Good paper - some comments to further improve the work in the future**

**Rating:** 7
**Confidence:** 4

**Review:**

This paper provides a study which evaluates the quality of synthetic image generation method from the lens of Turing Test. Specifically, it performed studies that involved radiologists examining both real and synthetic images to perform: (1) diagnostic task; and (2) real/fake identification task.

I think the study and the results presented in this paper are interesting and will offer very good contributions to the community.
Several comments for the authors to improve their work in the future:
1. Variation on the LDM. Currently the study only uses RoentGen model. I would suggest adding other existing models in the study to confirm that the observations hold for the general cases.
2. From Table 2 it looks like the variability in time to perform both tasks are quite high (e.g., VTT + Synthetic has variability as high its average). Could the authors provide explanations as to why the variability is high? is it caused by the diverse in the radiologist background (presented in Table 1)? If yes, it would be interesting to see the effect of years of experience with respect to the two tasks (e.g., would radiologists with shorter years of experience have less ability to differentiate between synthetic and real images?)
3. Due to high variability reported in Table 2, I would strongly suggest adding the std deviation in Table 3 too.
4. For writing, I would suggest group the metrics accordingly when they are first introduced in Results. Currently, the list of metrics are too long and it is not clear why they are used until the discussions get into subsection 4.1-4.5. Perhaps group them according to the questions we would like to investigate?

In overall, this paper is appropriate for the workshop and the presented results will provide good directions for the future work and how we can further improve LDM to produce realistic xray images.

---

### Official Review · Reviewer_2Pkx · 2025-03-27

**Rating:** 5
**Confidence:** 4

**Review:**

**Paper Summary**:

This paper introduces a practical framework, GazeVal that could exploit gaze patterns from experts of radiologists for quality assessment of synthetic medical images (X-ray images). Specifically, the authors propose to present a mixed sample set of real and synthetic X-ray images (generated by a conditional generative model) to radiologists for reviewing two tasks: diagnosis and synthetic image detection. During the process, the authors record their eye gaze and then use these gaze data and results of the review process to quantify the fidelity of synthetic images. Sixteen radiologists are involved in the experiments and results reveal that 96.6% of the generated images were identified as fake, which show limitation in high-fidelity image generation of medical images.

**Strengths**:

- The introduction is well-motivated by raising concerns about synthetic images being used in medical tasks.
- The visual illustration of the framework is straightforward and the paper is easy-to-follow.
- Detailed information about the participants as well as setup of the experiments are provided which clearly demonstrate the transparency of the research.

**Weaknesses**:

- The rationale behind choosing number of inference steps as 75 to generate X-ray images at line 179 is not well-discussed. Why should it be 75 inference steps while the standard denoising steps for LDM is 50 steps?
- The sample size for experiments are too small (only 30 for synthetic images and 30 for real images).
- The synthetic images used in the experiments are mostly generated by a specific generative model (LDM). Incorporation of synthetic images generated by other generative models should be considered to strengthen the claim made by authors.

In general,  this work presents an interesting study and observation about recent limitation of using synthetic images (X-Ray images) for medical tasks. I appreciate the author’s effort in providing detailed experimental settings to support the study. However, the experiments are still limited in terms of the experiment scale (small sample sizes), as well as diversity in synthetic images (generated by a particular generative model). Hence, I lean toward rejection.

---

### Decision · Program_Chairs · 2025-03-31

**Decision:**

Accept (Oral)

**Comment:**

The paper presents a new framework for evaluating the quality of synthetic medical images based on eye gaze tracking and direct diagnosis from radiologists. Two out of three reviewers lean positive in their reviews, emphasising the valuable insights the paper provides into the synthesis of medical images. The final recommendation is an acceptance. The authors are encouraged to strengthen the experiments by adding results from different diffusion models to understand the generalizability of the proposed framework.